# Ensemble-based Adversarial Defense Using Diversified Distance Mapping

## Abstract

We propose an ensemble-based defense against adversarial examples using distance map layers (DMLs). Similar to fully connected layers, DMLs can be used to output logits for a multi-class classification model. We show in this paper how DMLs can be deployed to prevent transferability of attacks across ensemble members by adapting pairwise (almost) orthogonal covariance matrices. We also illustrate how DMLs provide an efficient way to regularize the Lipschitz constant of the ensemble's member models, which further boosts the resulting robustness. Through empirical evaluations across multiple datasets and attack models, we demonstrate that the ensembles based on DMLs can achieve high benign accuracy while exhibiting robustness against adversarial attacks using multiple white-box techniques along with AutoAttack.

## 1 Introduction

Ongoing research has provided defenses against adversarial examples, which are crafted from correctly classified inputs with imperceptible perturbations. Despite the success of ensemble learning as a mechanism for reducing prediction errors and improving generalization by combining predictions of multiple models performing the same task (Russakovsky et al., 2015; Sagi & Rokach), early research has shown the ineffectiveness of mutiple ensemble-based defenses against adversarial examples and even has gone further to suggest that ensembles are only as robust as their weak components (He et al., 2017). If this claim is true, it basically defeats the purpose of using an ensemble, which is building a strong model out of weaker ones. Nevertheless, research continued to investigate the usage of ensembles as a defense mechanism (Pang et al., 2019; Verma & Swami; Sen et al.). However, recent attempts have been quickly shown ineffective (Tramèr et al.; Croce & Hein, 2020). We believe that one of the primary reasons for the weakness of ensemble defenses is the inter-model attack transferability. It was shown that even fundamentally different models could exhibit high attack transferability rate (Papernot et al., 2017; Kurakin et al., 2018). This phenomenon hindered the consideration of ensemble learning as a strong defense mechanism on its own. The reason is that if the member models are not robust, and they exhibit high attack transferability rate, attacks generated from one model can attack the rest, and hence, attack the entire ensemble.

In this work, we show that it is possible to circumvent the problem above and instill diversity among ensemble members via the employment of specially initialized and optimized distance-map-layers (DMLs), and hence throttle the inter-model attack transferability. Moreover, we demonstrate that DMLs provide spontaneous regularization of the Lipschitz constant, and therefore further boost the robustness.

The rest of this paper is organized as follows: We first discuss an overview of relevant recent works on ensemble defenses against adversarial examples and background information. Then we introduce a distance map layer based on Mahalanobis distance, and also explain the threat models considered in this work. We describe the creation of ensemble of DML-based individual models. Afterwards, we introduce a randomized version of DML. Finally, we evaluate the robustness of our ensemble model over MNIST, CIFAR-10 and RESISC-45 datasets.

## 2 BACKGROUND AND RELATED WORK

Not long ago, researchers revealed the vulnerability of machine learning models, particularly deep neural networks (DNNs), against adversarial examples (Szegedy et al., 2014). These models can provide incorrect predictions on examples that are slightly perturbed from correctly classified ones. The process of generating adversarial examples from natural ones is called *adversarial evasion attacks* (Biggio et al.). Evasion attacks can be categorized into black-box, gray-box, and white-box attacks. In the black-box setting, the attacker does not have access to the model's parameters and the potential defense of the model (Papernot et al., 2017; Brendel et al., 2018). In the gray-box setting, although the attacker does not have access to the model's parameter, it is aware of the defense applied in the model. In the white-box evasion, the attacker has access to both the parameters and the defense applied in the model. Therefore, the attacker can apply gradient-based attack techniques (Goodfellow et al., 2014; Carlini & Wagner, 2017; Madry et al., 2018), or it can even design a custom made adaptive attack based on its knowledge about the model and the defense (Tramèr et al.). From the defense perspective, white-box attacks are certainly the most challenging type of evasion attacks. Our proposed defense targets white-box attacks. Specifically, we focus on $L_p$ restricted perturbations of images, and, as it is typical in many research works, we consider the $L_\infty$ perturbations.

Since the discovery of adversarial attacks, a significant research activity has been devoted to developing appropriate defenses (Madry et al., 2018; Zhang et al.). However, most of the proposed defenses have been soon defeated (Tramèr et al.; Croce & Hein, 2020). An intuitive mechanism to defend against adversarial examples is training ensembles of other models in order to enhance their defenses (Pang et al., 2019; Verma & Swami; Sen et al.). However, the efficacy of existing ensemble approaches is often faced with skepticism due to the transferability phenomenon (Papernot et al., 2017), which makes the ensemble model believed to be at most as robust as its strongest constituent model (He et al., 2017). In this paper, we promote the diversity among individual networks from a new perspective via distance map layers. Our construction is based on the assumption that defending against transferred adversarial example is effective, even if the individual models are vulnerable to direct attacks. Our approach is orthogonal to the previous approaches and can be combined with other generic weak or strong defenses to further enhance the ensemble's resistance to adversaries.

Although there is no strict illustration on what is intuitively defined as diversity, in this work, we define the diversity as the highest dissimilarity between the learned features of different classes across the ensemble members. Diversity is greater when the prediction errors of individual members are highly uncorrelated (Liu & Yao, 1999a;b; Dietterich, 2000; Liu et al., 2019). This property may lead an adversarial perturbation to fail to fool the majority of networks in the ensemble. Topologically, diversity can be depicted as the variability in the shapes of the decision boundaries and inter-class neighborhood relationships in the embedding space. Changing the shape of the decision boundaries in the embedding space implies different loss shapes. With diversity in the shape of the losses corresponding to different models, the gradient from one model would not be a good approximation (transferable) to the other model. Note that due to the high capacity of DNNs, changing the topology of the embedding space can have no or negligible effect on the model's accuracy. Using this perspective on model diversity, we will illustrate that DML-based ensembles could be designed to achieve the requirement of diversity in prediction errors through injecting dissimilarities only between the DMLs of ensemble members. Previous works increase the diversity over training data via promoting the diversity of prediction errors of ensemble members (Liu & Yao, 1999a;b; Liu et al., 2019).

## 3 TECHNICAL APPROACH

In this section, we elaborate on our model with distance map layer (DML). We start by recalling the definition of Mahalanobis distance between two point $x_1, x_2 \in R^n$:

$$d_M(x_1, x_2) = \sqrt{(x_1 - x_2)^T M (x_1 - x_2)}, \tag{1}$$

where $M$ is the inverse of the covariance matrix (referred to as the $M$ matrix for the rest of the paper), which is a positive semi-definite matrix.

**Lemma 1.** *The Mahalanobis distance is $k$-Lipschitz continuous with $k = \sqrt{2} \|P\|_2$, where $M = P^T P$, and $P$ is a triangular matrix.*

A proof for this lemma can be found in (Zantedeschi et al., 2016). We now introduce a mapping function, based on the Mahalanobis distance, that can be used as a differentiable layer in a neural network. The function maps an input vector to an output vector by calculating Mahalanobis distances between the input and the learned centers of the function. Equation 2 shows the element of the output vector corresponding to center $c$ for an input vector $x$. We call this function *distance map layer (DML)*.

$$d_{M,c}(x) = \sqrt{(x-c)^T M (x-c)}. \tag{2}$$

In this formula, $M \in \mathbb{R}^{N \times N}$ and $c \in \mathbb{R}^N$ are the learnable parameters of the DML. When the DML is used as the penultimate layer of a classification neural network (i.e. the layer producing logits), each center will correspond to one class. Essentially, the layer maps an input vector to a vector of distances to class centers. Note that, in this usage, the negative of the distances has to be used to maintain the typical assumption of DNNs that the highest logit corresponds to the most likely class. The classification loss function will encourage the absolute distance to the target class to be small while the distances to all other classes to be large, which induces compactness in the embedding space. This compactness has been used in prior work for classification problems (Wan et al., 2018) and for adversarial robustness of individual models (Pang et al., 2020).

Our setting is composed of combining a classification network $G$ with a DML $h$ to obtain a model $F$ characterized by $F(x) = h \circ G(x)$ where $h : \mathbb{R}^M \to \mathbb{R}^M$ and $G : \mathbb{R}^N \to \mathbb{R}^M$. In other words, $G$ is used as a prior to map the input space to a representation space, and $h$ maps the representation space to distances from the corresponding class centers. In the next section, we demonstrate how a distribution shift that $F$ shows with respect to $G$ could be efficiently utilized to form a diverse ensemble of models. Before that, we want to highlight an important feature about DML. If the network $G$ is $L_G$-Lipschitz continuous, and the DML $h$ is $L_h$-Lipschitz continuous, then the network $F$ is Lipschitz continuous with constant upper-bounded by $L_G \cdot L_h$. Thus, imposing small $L_h$ during the training process may lead to the regularization of Lipschitz constant for the network $F$. It is shown in the appendix that this can also largely improve the certification bounds of network $F$ compared to the model $G$.

### 3.1 ENHANCING DIVERSITY BETWEEN ENSEMBLE MEMBERS

The main part of the proposed ensemble model is the deployment of DMLs. The ensemble members stacked with DMLs are trained interactively to achieve the highest level of diversity. In particular, a DML ensemble is trained to satisfy these properties: 1) the ensemble members have the highest accuracy over benign samples and diverse prediction errors over adversarial samples; 2) the attacks generated on one ensemble member are not transferable to other ensemble members; and 3) the training procedure maintains the Lipschitz constant of an ensemble member model small. The first objective is important as it indicates that the ensemble could provide high accuracy even with the low accuracy of the individual members. The second objective guarantees a well-performing ensemble model under evasion attacks. That is an adversarial example crafted on an individual ensemble member model would normally be misclassified in that individual member, but not necessarily in the other ensemble members. This goal can be achieved in two folds: 1) by randomly choosing the centers and shuffling them to vary over classifiers in the formulation of DML; 2) by imposing the $M$ matrices to be dissimilar and possibly orthogonal between the ensemble members. Increasing the cosine dissimilarity of DML's covariance matrices of ensemble members would lead to nontransferable adversarial perturbations (Adam et al., 2019). One approach to reduce the transferability across several models is to enforce various geometric relationship like orthogonality between the input-output gradient of the models. For two distance map layers $f$ and $g$ with corresponding $M_f$ and $M_g$ inverse covariance matrices, we have shown in the appendix that the orthogonality of the $M$ matrices, i.e., $M_f M_g = 0$ is a sufficient condition for the input-output gradients of $f$ and $g$ to be orthogonal. Note that the explored sufficient condition is independent of the input $x$. Roughly speaking, the increase in the dissimilarity of the $M$ matrices reduces the cosine similarity of input-output gradients of the combined network by the chain rule formula. Therefore, practically, minimizing the cosine similarity of the $M$ matrices across models enables reducing the rate of transferability of adversarial attacks across models in the ensemble.

## 3.2 DML PARAMETER INITIALIZATION AND TRAINING

In this section, we describe the procedure of simultaneous and interactive training of ensemble members towards the goal of enhancing diversity. The main building block of various ensemble members using DML is that the centers and the inverse covariance matrices of members are trained to have the highest level of dissimilarities between the ensemble members. We aim to increase the failure independence by enhancing the disagreement diversity of different networks. We recall that these ensemble members are generated using a reference model by stacking a DML to the last hidden layer of the prior model. The ensemble model consists of the created DML-based models. To train the ensemble members, the parameters of each model learned to increase the dissimilarity of the $M$ matrices (e.g., by decreasing the dissimilarity loss function). The DML's centers in the ensemble are shuffled across its members. In practice, we initialize the centers to be a permutation of the rows of an identity matrix. Applying the shuffled normal vectors as the centers of the DML would avoid the scaling of the gradient of the network and can decrease the risk of gradient masking. Shuffling the centers across the individual members may cause to diversity in adversarial directions for the members. In this way, the ensemble framework can potentially be used to detect adversarial examples. We discuss more about this mechanism in appendix. Dissimilar $M$ matrices in the ensemble of models alternate the way in which each individual learner traverses the hypothesis space. We also make the $M$ matrix partially learnable, i.e., only certain elements in $M$ are trainable and the rest are fixed during the training. This would make the classifiers have different prediction errors on different labels. In the following, we elaborate on how we split the learnable parameters of the $M$ matrix across members to encourage diversity through diverse tasks.

## 3.3 META ENSEMBLE MODEL

We choose the trainable parameters in $M$ such that there would not be any overlap between the indices of the learnable elements of different ensemble members. More specifically, let $I_j$ denote the learnable parameters of the $M$ matrix of the DML in the $j^{th}$ ensemble member. To promote the diversity in the topology of the DNNs, we select $I_j$ such that $I_j \cap I_k = \emptyset$ for $j, k \in [K]$. By allowing a classifier to learn specific elements in $M$, it causes the model to be more accurate on the classes with labels corresponding to those elements, e.g., the $k^{th}$ ensemble model would show better benign accuracy on the labels in the set $I_k$. On the other hand, during experiments, we realized that the classifiers are more vulnerable on those labels when exposed to the attackers. Since the classifiers are more susceptible on those labels, the predictions from ensemble members could be aggregated while excluding the predictions of each member classifier's specific labels, e.g., $I_k$ for classifier $k$. Due to the fact that the sets $I_k$'s for $k \in K$ have no pairwise intersections, neglecting the prediction of models on those particular labels when aggregating the final prediction would not impose any artificial bias on the ensemble. We will refer to the aforementioned mechanism in the following as meta-model. In summary, we are training models which are fundamentally diverse in their robustness to different adversarial attacks, and therefore it increases the varieties in the shape of decision boundaries and consequently adversarial directions in the ensemble.

We use a tailored loss to interactively train the ensemble members. Formally, the loss function is formulated as

$$\mathcal{L} = \sum_{k \in [K]} \mathcal{L}_{CE}^k + \gamma \sum_{k \in [K]} \sum_{t \in [K] \wedge t \neq k} \|M_k \cdot M_t\| + \beta \sum_{k \in [K]} \|M_{I_k}\|_1 \qquad (3)$$

where $K$ is the number of ensemble members, $L$ is the number of classes, $\mathcal{L}_{CE}^k$ is the classification loss (cross entropy) of $k^{th}$ ensemble member.

Minimizing the sum of losses from ensemble members increases the benign accuracy of the ensemble model, while the other terms in the loss formula (Equation (3)) manipulate the arrangement of the decision boundaries. The second term encourages the dissimilarity of covariance matrices and thus advocates diverse hypothesis for classifiers. The third term is a regularization parameter for the sum of $L_1$ norm of DML's $M$ matrices to achieve a DML with a low Lipschitz constant. The positive real numbers $\gamma$ and $\beta$ balance these terms with the cross-entropy objective. We choose these parameters via hyper-parameter tuning. For training, one needs to initialize the centers and the $M$ matrices of the DML in each classifier to have the most dissimilarity to the other ensemble members. Towards

having a DML with a low Lipschitz constant, we consider a diagonal $M$ matrix, where the diagonal terms are initialized randomly from the uniform distribution between 0 and 1. It is possible to opt not to train the $M$ matrices if there exist sufficient dissimilarities between paired matrices (i.e., orthogonality) via appropriate initialization, given that networks trained from different initialization can be far apart in their internal representations.

Finally, a consensus method over the predictions from individual classifiers is performed to obtain the output of the ensemble model. The consensus method could be any operation on the prediction outputs, e.g., majority voting, mean, median, sum. In this work, the ensemble output is based on the majority voting of $k$-top predictions of the ensemble members.

## 4 Randomized DML and uncertainty in the DML-based DNN

Considering the distance map layer $f(x) = \sqrt{(x-c)^T M (x-c)}$, where $M = P^T P$ and $P$ is a lower-triangular matrix, we construct a randomized version of DML $g$ from the base DML $f$. The randomized DML $g$, is the expectation of the isotropic Gaussian perturbation of the $M$

$$g(x) = \mathbb{E}_{\delta \sim \mathcal{N}(0,\sigma^2 I)} \left[ \sqrt{(x-c)^T (P+\delta)^T (P+\delta)(x-c)} \right], \tag{4}$$

where $\sigma$ is a hyper-parameter.

Here, we apply the uncertainty of a Gaussian process to identify the low-confidence regions of the input space to the DML layer. The Gaussian process assumes a set of priors on the set of all functions that can map the input space to the output space. At training time, for any sample $x$ with target classification $y$, only the function of the center corresponding to $y$ will be considered. At inference time, the expectation of all the output of these functions are used as the prediction. As the $M$ matrix determines the shape of decision boundaries, uncertainty estimation from randomized smoothing of $M$ can be extracted during the inference time. We sample i.i.d. Gaussian samples $\delta_1, \ldots, \delta_T \sim \mathcal{N}(0, \sigma^2 I)$, and use the following Monte-Carlo estimator for the expectation of the randomized DML $g$

$$g(x) \approx \frac{1}{T} \sum_{i=1}^{T} \sqrt{(x-c)^T (P+\delta_i)^T (P+\delta_i)(x-c)}. \tag{5}$$

In the experiments, we show that the randomized DML provides better robust accuracy compared to DML when used in an ensemble.

## 5 Experiments

In this section, we present our empirical studies to show the effectiveness of our proposed method on decreasing the threat from adversarial attacks, while showing high benign accuracy. We perform our experiments on MNIST, CIFAR-10 and RESISC-45 (Cheng et al., 2017) detasets. Each of MNIST and CIFAR-10 has 10 classes and RESISC-45 has 45 classes. We implement the ensemble model based on the networks shown in Table 1. In the experiments for each dataset, we consider an ensemble of five models. Our baseline model is the ensemble of five models without DML trained with cross-entropy loss. The DML-based ensemble models are jointly trained based on the loss in Equation (3). We set $\beta = 0.01$ and $\gamma = 0.01$ in Equation (3) for training the ensemble models over all three datasets. Following the illustration of the ensemble meta-model, for each classifier, only two elements of the diagonal elements of the $M$ matrix are trainable and the rest of the diagonal elements are set to the fixed value 0.1. For all the individual models in the ensemble and for all datasets, the center points of DMLs are set to the columns of an identity matrix with different shuffling across members. The centers are fixed during the training procedure.

In Figure 1, we show the distribution of the positive CLEVER score values (Weng et al., 2018) for 1000 randomly selected images from the MNIST dataset for a model formed by summing the outputs of ensemble members. We used the $L_2$ radius of 2 to calculate the CLEVER scores. The results for the other datasets, which is similer to MNIST's, are provided in the appendix. The results shows that the number of samples with non-zero CLEVER score is an order of magnitude higher than the counterpart for the baseline model. Moreover, for the baseline model, there exist no sample that has a

| Dataset | MNIST | CIFAR-10 | RESISC-45 |
|---------|-------|----------|-----------|
| Model | LeNet (Lecun et al., 1998) | ResNet-18 (He et al., 2016) | |

Table 1: Base models for creating DML-based models for each dataset

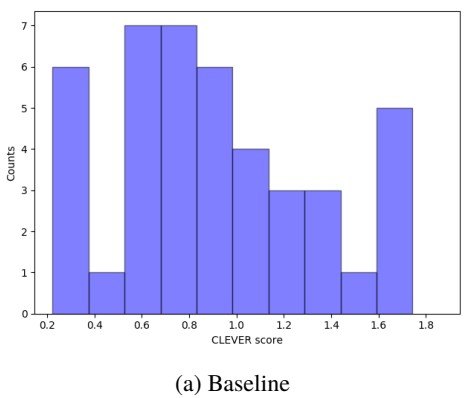

(a) Baseline

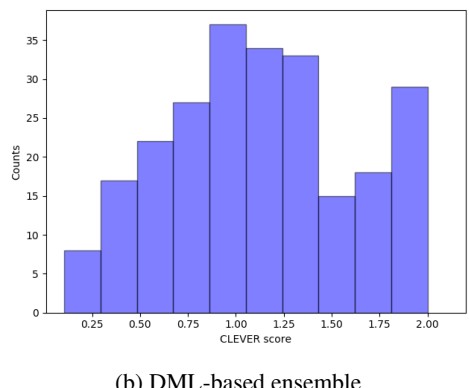

(b) DML-based ensemble

Figure 1: Distribution of non-zero CLEVER scores for baseline ensemble and DML-based ensemble for 1000 randomly selected images from the MNIST dataset. The CLEVER scores are computed on the $L_2$-norm ball with radius 2.

score higher than 1.7 while for DML, the highest score is close to 2. This shows that the DML-based ensemble has an intrinsic boost in robustness compared to the baseline.

In Tables 2-3, we show the benign and adversarial accuracy of ensemble models over the three datasets. The white-box attacks that we consider are FGSM (Goodfellow et al., 2014), BIM (Kurakin et al., 2017), PGD (Madry et al., 2018), and Carlini & Wagner (CW) (Carlini & Wagner, 2017), which are commonly used for evaluating defense models. The number of iterations for BIM and PGD are set to 10, while for CW, it is set to 200. The $L_\infty$ budget $\epsilon$ for FGSM, PGD, and BIM; and the penalty (initial constant) parameters $\gamma$ for CW are shown in the table. The attacks we consider in this section are untargeted attacks unless otherwise mentioned. In untargeted attacks adversaries are constructed to cause the classifier to produce any incorrect label. Unlike the previous research on ensemble-based defenses (Sharif et al., 2019; Liu et al., 2019) where the adversaries were crafted over a target model, in this paper, the adversarial examples are generated over the ensemble model derived by summing the predictions. Crafting adversarial examples over the ensemble model would assess the true resistance of the ensemble as the generated adversaries contain the gradient information from all the individual members. The predicted labels for the benign and adversarial samples in the ensemble model are obtained using the majority votes of top-1 predictions of the individual classifiers. The results show a remarkable performance enhancement over the baseline model. This could be explained by high non-transferability of adversaries and diverse ensemble representation across the ensemble members, which hinders the process of finding an optimal adversary that could fool the majority of ensemble members to have the correct prediction changed.

Table 2: MNIST white-box attack results. FGSM and PGD are applied with specific perturbation budget $\epsilon$. CW is used with the specified penalty (initial constant) parameter $\gamma$.

| **Training Model** | Benign Acc | FGSM $\epsilon = 0.2$ | FGSM $\epsilon = 0.3$ | BIM-10 $\epsilon = 0.2$ | PGD-10 $\epsilon = 0.2$ | CW $\gamma = 1$ |
|---------|-------|------|------|------|------|------|
| Baseline | 98.79 | 23.81 | 3.12 | 0.00 | 0.00 | 0.00 |
| DML Ensemble | 93.66 | 68.55 | 39.62 | 63.08 | 62.92 | 85.00 |

Table 3: White-box attack results for CIFAR-10 and RESISC-45 datasets. FGSM and PGD are applied with specific perturbation budget $\epsilon$. CW is used with the specified penalty (initial constant) parameter $\gamma$.

| Training Model | Benign Acc | FGSM $\epsilon = 0.015$ | FGSM $\epsilon = 0.031$ | BIM-10 $\epsilon = 0.015$ | PGD-10 $\epsilon = 0.015$ | CW $\gamma = 1$ |
|---|---|---|---|---|---|---|
| **CIFAR-10** | | | | | | |
| Baseline | 85.66 | 23.84 | 6.15 | 7.75 | 6.15 | 27.67 |
| DML Ensemble | 84.57 | 56.61 | 43.35 | 38.35 | 39.87 | 39.54 |
| **RESISC-45** | | | | | | |
| Baseline | 85.66 | 3.75 | 1.34 | 0.00 | 0.00 | 45.22 |
| DML Ensemble | 87.28 | 52.06 | 39.64 | 45.07 | 45.12 | 61.84 |

Table 4: Adversarial accuracy of all ensemble models when the attacks are crafted on **Model 1** with PGD-10. The labels for the targeted attacks are selected randomly and different from the true labels. The size of perturbation $\epsilon$ is 0.2, 0.015, 0.015 for MNIST, CIFAR-10, and RESISC-45, respectively.

| Dataset | Accuracy | Ensemble | Model 1 | Model 2 | Model 3 | Model 4 | Model 5 |
|---|---|---|---|---|---|---|---|
| **MNIST** | Benign | 93.66 | 86.56 | 69.57 | 95.40 | 93.35 | 95.36 |
| | Untargeted | 58.24 | 33.79 | 44.96 | 66.14 | 53.99 | 63.83 |
| | Targeted | 83.07 | 58.24 | 58.03 | 84.53 | 76.49 | 83.64 |
| **CIFAR-10** | Benign | 84.57 | 82.08 | 82.02 | 81.30 | 81.42 | 81.64 |
| | Untargeted | 61.35 | 40.93 | 64.61 | 64.15 | 63.40 | 63.84 |
| | Targeted | 68.60 | 48.45 | 70.51 | 69.12 | 69.07 | 69.42 |
| **RESISC-45** | Benign | 87.29 | 75.33 | 78.33 | 79.67 | 79.51 | 79.89 |
| | Untargeted | 74.02 | 30.80 | 68.44 | 69.42 | 71.07 | 69.31 |
| | Targeted | 75.33 | 35.58 | 72.89 | 74.40 | 75.42 | 74.12 |

We now examine if the adversaries generated on one ensemble member could be transferred to the other models in the ensemble. For this purpose, in Table 4, we provide the results for both targeted and untargeted attacks when the crafted attacks on Model 1 are transferred to the other constituent ensemble members and the ensemble model. It is observed that the crafted attacks exhibit low transferability rate. This could be illustrated by the visualisation of the loss function around a sample from RESISC-45 dataset in Figure 2, which shows that the shapes of the loss surfaces of the constituent models are diverse and therefore the input-output gradients of a model do not provide a close approximation for the other models included in the ensemble.

In Table 5, we compare the distortion needed for the ensemble model versus the individual members for the targeted attacks using adaptive CW-$L_2$ attack (He et al., 2017) for 100 randomly selected samples from each dataset. The results show that the perturbation required to generate adversaries on individual member is multiple times lower than the distortion required to create adversaries on the ensemble models.

Finally, in Table 6 we compared the robust accuracy for DML-ensemble and randomized DML ensemble over CIFAR-10 datasets. The results for the other dataset could be found in appendix. The results from randomized DML-ensemble shows outperform adversarial accuracy.

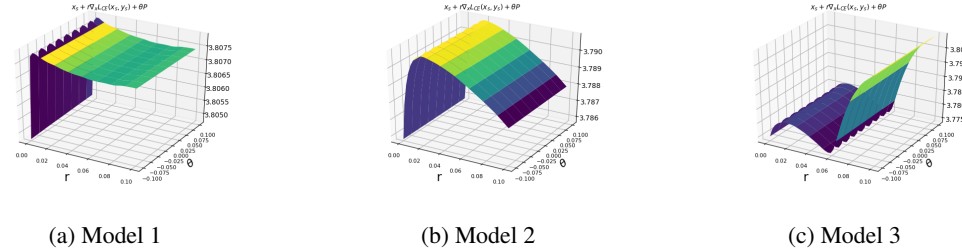

| (a) Model 1 | (b) Model 2 | (c) Model 3 |
|---|---|---|

Figure 2: Loss surface for individual classifiers around a sample $x_s$ from RESISC-45 dataset for points $x_s + r\nabla_x L_{CE}(x_s, y_s) + \theta P$ with $P \sim \mathcal{N}(0, 0.05)$, $r \in [0, 0.1]$, and $\theta \in [-0.1, 0.1]$.

Table 5: The required $L_2$ distortion for adaptive CW-$L_2$ attack (He et al., 2017) with $\gamma = 1$ for the combined and individual ensemble members of size 5 for 100 randomly selected images for each dataset. The number of iterations and stepsize are 1000, 0.1, respectively. $\kappa$ is denoting the confidence parameter for CW-$L_2$ attack.

| Dataset | $L_2$ distortion required for generating adversaries on each model | | | | | | |
|---|---|---|---|---|---|---|---|
| | Ensemble $\kappa = 0$ | Ensemble $\kappa = 0.1$ | Model 1 | Model 2 | Model 3 | Model 4 | Model 5 |
| **MNIST** | | | | | | | |
| Adversarial accuracy | 66.00 | 45.00 | 0.00 | 8.00 | 1.00 | 1.00 | 4.00 |
| Average distortion | 2.31 | 3.79 | 1.34 | 1.41 | 1.52 | 1.36 | 1.52 |
| **CIFAR-10** | | | | | | | |
| Adversarial accuracy | 52.00 | 12.00 | 0.00 | 0.00 | 0.00 | 0.00 | 0.00 |
| Average distortion | 0.46 | 23.30 | 0.52 | 2.67 | 2.69 | 0.51 | 0.45 |
| **RESISC-45** | | | | | | | |
| Adversarial accuracy | 73.00 | 71.00 | 0.00 | 0.00 | 0.00 | 0.00 | 0.00 |
| Average distortion | 3.60 | 6.24 | 1.47 | 1.63 | 2.06 | 2.36 | 2.53 |

Table 6: Accuracy of white box attacks on DML-based ensemble and randomized DML (R-DML) ensemble (R-DML-ensemble) for the perturbation budget $\epsilon$ specified for each dataset. The attacks are performed with 5 random restarts.

| Dataset | Benign Acc | FGSM | PGD-20 | PGD-40 |
|---|---|---|---|---|
| **CIFAR-10** ($\epsilon = 8/255$) | | | | |
| DML Ensemble | 84.57 | 47.72 | 19.38 | 14.46 |
| R-DML Ensemble | 81.25 | 56.48 | 38.12 | 37.34 |

Table 7: MNIST AutoAttack (Croce & Hein, 2020) results. The adversaries are crafted with the specified perturbation budgets $\epsilon$.

| Training Model | Benign Acc | AutoAttack | | |
|---|---|---|---|---|
| | | $\epsilon = 0.1$ | $\epsilon = 0.2$ | $\epsilon = 0.3$ |
| DML Ensemble | 93.66 | 88.47 | 73.46 | 49.44 |

## 6 AutoAttack on the DML-based ensemble

In this section, we evaluate our proposed model by the ensemble of parameter-free attacks, AutoAttack (Croce & Hein, 2020). In (Croce & Hein, 2020), several recently introduced defenses, including the ensemble model in (Pang et al., 2019), were evaluated on AutoAttack, and most of them were shown to be, at best, less robust than they had been thought to be according to their original papers. For instance for the ensemble defense proposed in (Pang et al., 2019), it was reported that the accuracy of the model dropped to zero under AutoAttack on CIFAR-10. In Tables 7-8, we show the results for the evaluation of our proposed model under AutoAttack. The results indicate that the DML-based ensemble models exhibit notable adversarial robustness.

## 7 Conclusion

In this paper, we introduces new Mahalanobis distance based ensemble models, in which the properties of the distance function are leveraged to increase the diversity of the ensemble predictions and reduce the attack transferability rate across the individual members. The ensemble members initialized in a way that induces diversity, and are jointly trained while encouraging orthogonality among the inverse covariance matrices of their distance functions, which is shown to reduce transferability. The proposed mechanism can be applied to pretrained networks as priors to form an ensemble with enhanced properties, which makes our model complementary to other existing defenses. Our results show that the proposed ensemble-based defense achieves high benign accuracy and adversarial accuracy across multiple attack algorithms, including AutoAttack, without using adversarial training.

Table 8: AutoAttack (Croce & Hein, 2020) results for CIFAR-10 and RESISC-45 datasets. The adversaries are crafted with the specified perturbation budgets $\epsilon$.

| Training Model | Benign Acc | AutoAttack | | |
|---|---|---|---|---|
| | | $\epsilon = 4/255$ | $\epsilon = 8/255$ | $\epsilon = 16/255$ |
| **CIFAR-10** | | | | |
| DML Ensemble | 81.62 | 40.23 | 16.8 | 6.77 |
| **RESISC-45** | | | | |
| DML Ensemble | 87.28 | 63.20 | 48.53 | 26.28 |

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

## A  APPENDIX

### A.1  CERTIFIABLE DML

Distance-map layer is a nonlinear map composed of square root of a quadratic function. We consider the distance-map function $h$

$$h(x) = \sqrt{(x - c)' M_h (x - c)}$$

where $M_h$ is a semi-definite positive matrix with a Choleskey decomposition denoted by $M_h = L'_M L_M$. We let $\phi^{r-1}$ denote the input to the layer $h$. Then, we obtain

$$
\begin{aligned}
h(x) = \sqrt{(\phi^{r-1} - c)' M_h (\phi^{r-1} - c)} &= \sqrt{(\phi^{r-1} - c)' L'_M L_M (\phi^{r-1} - c)} \\
&= \left\| L_M (\phi^{r-1} - c) \right\| \leq \|L_M\| \left\| (\phi^{r-1} - c) \right\| \\
&\leq \left\| L_M \phi^{r-1} \right\| + \|L_M c\| = \left\| A_h \phi^{r-1} \right\| + B_h
\end{aligned}
\tag{6}
$$

On the other hand,

$$
\begin{aligned}
h(\phi^{r-1}) = \sqrt{(\phi^{r-1} - c)' M_h (\phi^{r-1} - c)} &= \sqrt{(\phi^{r-1} - c)' L'_M L_M (\phi^{r-1} - c)} \\
&= \left\| L_M (\phi^{r-1} - c) \right\| \geq \left\| L_M \phi^{r-1} \right\| - \|L_M c\| = \left\| A_h \phi^{r-1} \right\| - B_h
\end{aligned}
\tag{7}
$$

Thus, the certifiable bound for $h$ is given by

$$
\left\| A_h \phi^{r-1} \right\| - B_h \leq h(\phi^{r-1}) \leq \left\| A_h \phi^{r-1} \right\| + B_h
$$

If $M_h$ is a diagonal matrix, with $\alpha_M = \max_{i,j} |M_{ij}|$ and $\beta_M = \min_{i,j} |M_{ij}|$

$$
\beta_M^{\frac{1}{4}} \left\| \phi^{r-1} \right\| - B_h \leq h(\phi^{r-1}) \leq \alpha_M^{\frac{1}{4}} \left\| \phi^{r-1} \right\| + B_h
$$

## A.2 ORTHOGONALITY OF INPUT-OUTPUT GRADIENTS OF PAIRED DMLS

One approach to reduce the transferability across several models is to enforce various geometric relationship like orthogonality between the input-output gradient of the models. We consider two (squared) distance map layers $f$, and $g$ defined by $f(x) = (x - c_f)' M_f (x - c_f)$ and $g(x) = (x - c_g)' M_g (x - c_g)$, where $M_f$ and $M_g$ are the $M$ matrices, and $c_f$ and $c_g$ are centers, corresponding to $f$ and $g$. We have

$$
\nabla_x f(x) = \frac{M_f x - M_f c_f}{f(x)}
$$

and

$$
\nabla_x g(x) = \frac{M_g x - M_g c_g}{g(x)}
$$

Thus, we obtain

$$
\nabla_x f(x) \cdot \nabla_x g(x) = \frac{1}{f(x) g(x)} (x' M_f M_g x - x' M_f M_g c_g - c'_f M_f M_g x + c'_f M_f M_g c_g)
$$

where we used the fact that $M_f$ and $M_g$ are symmetric matrices. Given $M_f M_g = 0$, we obtain

$$
\nabla_x f(x) \cdot \nabla_x g(x) = 0.
$$

Thus, the orthogonality of the $M$ matrices for $f$ and $g$ is a sufficient condition for the input-output gradients of $f$ and $g$ to be orthogonal.

## A.3 THE THREAT MODEL FOR THE ENSEMBLE

To classify the input $x$ using an ensemble of classifiers, the output of each ensemble member on input $x$ is computed. The ensemble prediction is performed by applying a consensus method to integrate the predictions form all individual members. The consensus method could be like majority voting, mean, median, sum. If all the members can predict with uncorrelated error, then a simple averaging may reduce the prediction error by the number of members. However, when the errors are not independent, the reduction error would be smaller than this factor. In this work, the ensemble output is obtained by using the majority voting of $k$-top predictions of the ensemble members. In the experiments, since the majority voting operator is non-differentiable, the white box adversaries are generated by the ensemble model which is obtained via summing of the predictions and attacking

the aggregation layer of the ensemble. Unlike the previous research (Liu et al., 2019) where the attacks are generated on a target model which is either a constituent ensemble member or a model not included in the ensemble, the white box attacks in this work are generated on the ensemble layer. Generating the attacks on a target model might provide a false sense of robustness for the ensemble, while generating the adversaries by attacking the aggregation layer of the ensemble provides the strongest white-box attack having the gradient information from all the individual members. Thus, it can measure the real robustness of the ensemble model. We also investigate adaptive attacks (He et al., 2017) by crafting adversaries on all the component models of the ensemble. We show that the perturbation which is required to generate adversaries on individual member is by order of magnitudes lower than the distortion required to create adversaries on the ensemble models.

## A.4  DML-based ensemble for adversary detection

Given an input $x$, DML-based ensemble might be utilized for detecting adversaries. More particularly if the majority number of ensemble members which is determined by the threshold $v$ do agree on a class of the input class, the ensemble classifies the input as the majority class. Otherwise the input is identified as adversarial example. To elaborate, we let $f_k(x)$ is the output of the $k^{th}$ classifier on the input $x$ and $l = \arg\max_{i \in [L]} f_{ki}(x)$. Then the output of ensemble classifier $f_{ens}$ on $x$ is given by

$$f_{ens}(x) = \begin{cases} l & \text{if } \left|\{k \mid \arg\max_{i \in [L]} f_{ki}(x) = l, k \in [K]\}\right| \geq v \\ \text{adversary} & \text{otherwise.} \end{cases} \quad (8)$$

Evidently, increasing the threshold $v$ rises the probability of detecting adversaries, while the number of false positive cases might grows at the same time and therefore the benign accuracy of the ensemble model decreases. Practically, the threshold $v$ in (8) is selected as $\lfloor \frac{n}{2} \rfloor + 1$. If $v = K$ (i.e., the number of ensemble members), the majority framework is called unanimity framework were the ensemble classifies the input $x$ only if all the classifiers agree on the class of input $x$. Unanimity framework poses a security risk if the adversarial examples cause the misclassification to the same class over models. Otherwise a defender can use the disagreement between models as a mechanism to detect adversarial examples. An ensemble with diverse DML-based members is less likely to mislcassify the adversary input $x$ to the same class. Thus, the detector becomes more powerful when the predictions from constituent members of ensemble are diverse. Due to the shuffled representation of neighbour classes' domain to the decision region of class $x$ for the DML-based models, the adversaries for the sample $x$ is misclassified diversely (i.e., diverse labels) across the ensemble members. So the likelihood of two classifiers to misclassify the adversary input to the same class would decrease and it increases the probability to detect $x$. To insure the diversity in the centers form DML layer of the ensemble members, we shuffle the centers of classes per each individual classifier. Note that the diversity does not necessarily change the shape of the decision boundaries and only alter the adversarial directions. In the other words, it could cause the classifiers to provide various prediction on a given input that could be leveraged to detect adversaries. It is important to note that, unlike previous approaches on ensemble model (Sharif et al., 2019), imposing this diversity does not imply any upper bound in the number of constituent classifiers in the ensemble as there are infinitely many ways to have the centers of classifiers shuffled. In previous research (Sharif et al., 2019; Liu et al., 2019), the authors had to split the classifiers to plenty of ensembles in order to achieve diversity inside each ensemble.

### A.4.1  Unanimity framework for adversarial detection

We might use the ensemble of DML-based networks to detect adversarial samples based on unanimity framework. In this setting, the ensemble provides the prediction for label of input $x$, only if all the individual members agree on the label of the $x$. Otherwise, the ensemble renders the sample as an adversarial example. In Figure 3 we have shown the ROC plots and AUC scores for adversarial detection using unanimity framework over three datasets. For each dataset we considered testset of correctly classified benign samples and the corresponding adversarial samples crafted with white-box attacks FGSM and PGD. The parameters for the attacks used are similar to the ones in Tables 2-10. The results show the remarkable performance of DML-based ensemble with unanimity framework as a detection metric.

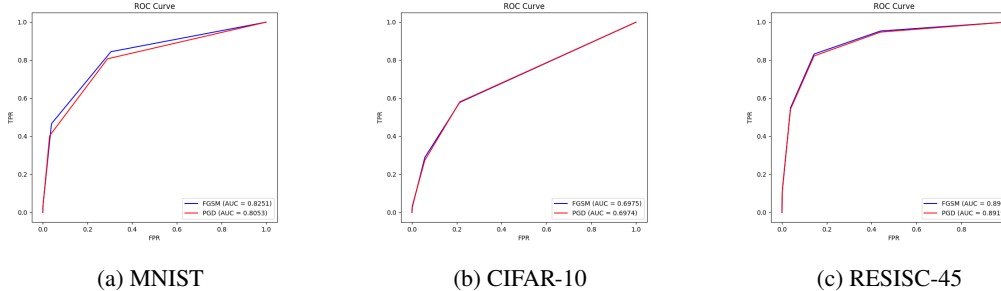

(a) MNIST  (b) CIFAR-10  (c) RESISC-45

Figure 3: ROC curves for adversary detection using unanimity framework. The perturbation size $\epsilon$ is 0.3, 8.0/255 and 8.0/255 for MNIST, CIFAR-10 and RESISC-45 datasets, respectively.

## A.5  ATTACK DETECTION MECHANISM WITH 1-D PROJECTION

Shuffling the centers in high dimensional space does not necessarily lead to diverse adversarial directions as in high dimension the decision boundaries of one class to the other class could be numerous. To circumvent this issue to promote the diversity of the adversarial direction by taking advantage of shuffled centers, we restrict the embedding space to a one dimensional centers, where the center location for each class per each model in the classifier is just a shuffled representation of fixed vector providing the center locations. In Figure 4 it is observed that shuffling the centers in 1-D can significantly increase the rate of adversarial detection.

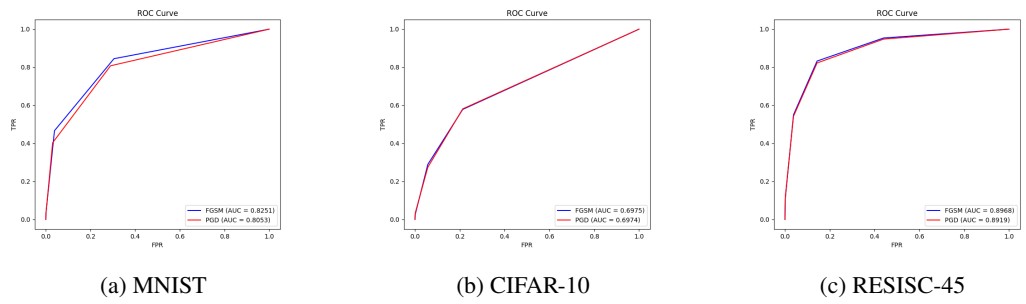

(a) MNIST  (b) CIFAR-10  (c) RESISC-45

Figure 4: ROC curves for adversary detection using unanimity framework with 1-D mapping in the embedding space. The perturbation size $\epsilon$ is 0.3, 8.0/255 and 8.0/255 for MNIST, CIFAR-10 and RESISC-45 datasets, respectively.

### A.5.1  ADVERSARIAL DETECTION USING UNCERTAINTY METRIC

We define the uncertainty distribution of the model at sample points as the variance of outputs of models across ensemble members and for multiple inferences. The quantified uncertainty of the randomized-smoothing DML-based network is utilized to detect adversarial examples. Since there is one output for each class in the network, the mean of the uncertainties for all classes indicates the uncertainty of the model at the sample $x$. In Figure 5 we have shown the uncertainty distribution of FGSM and PGD adversarial examples versus normal examples. Apparently, the distribution from adversarial examples is distinguishable from normal and noisy samples. We use this intuition to train a logistic-regression model on the inputs of uncertainty of data to detect adversarial examples.

In Table 9 we show the adversarial accuracy of the same ensemble models we considered in Tables 2-3 for the randomized smoothing DML layer indicated in equation (4). To compute the randomized smoothing DML layer with Monte-Carlo approximation formulation (5), we set $T = 20$ and the noises are generated from the normal distribution $\mathcal{N}(0, \sigma^2 I)$ with $\sigma = 0.1$. It is seen in Table 9 that the adversarial accuracies of white box attacks for randomized smoothing DML outperform the corresponding results for ensemble of DML based models.

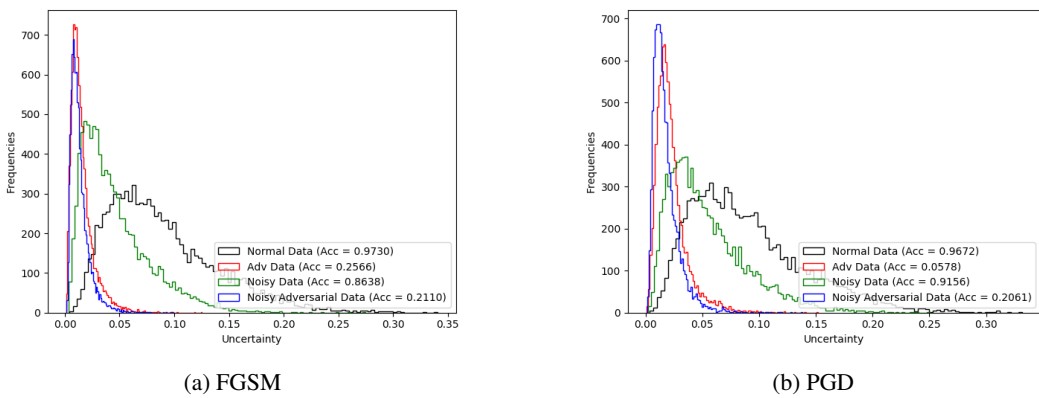

(a) FGSM

(b) PGD

Figure 5: Uncertainty distribution.

Table 9: Accuracy of extended white box attacks on DML based ensemble (DML-ensemble) and randomized smoothing DML ensemble (R-DML-ensemble) for the perturbation budget $\epsilon$ specified for each dataset. The attacks are performed with 5 random restarts.

| Dataset | Benign Acc | FGSM | PGD-20 | PGD-40 |
|---|---|---|---|---|
| **MNIST** ($\epsilon = 0.3$) | | | | |
| DML-ensemble | 93.66 | 42.04 | 36.18 | 35.37 |
| R-DML-ensemble | 93.50 | 44.98 | 42.07 | 41.73 |
| **CIFAR-10** ($\epsilon = 8/255$) | | | | |
| DML-ensemble | 84.57 | 47.72 | 19.38 | 14.46 |
| R-DML-ensemble | 81.25 | 56.48 | 38.12 | 37.34 |

We could also detect adversaries using the uncertainty quantity measured using randomized DML. As shown, the benign, noisy and adversarial data are representing different uncertainty distribution. Based on this observation, we train a logistic-regression classifier from the uncertainties of a combined benign and noisy data as the negative class and the uncertainty from adversarial data as the positive class. We use the testset of correctly classified samples for benign samples, and crafting noisy, and adversarial examples. The adversaries are generated using from the FGSM and PGD attackers. The trained model classifies benign and adversarial examples. We show the ROC plots and AUC scores of the classifier in Figure 6. The ROC plots demonstrate that the uncertainty from randomized DML might be used as an indicator to detect if a sample is adversary.

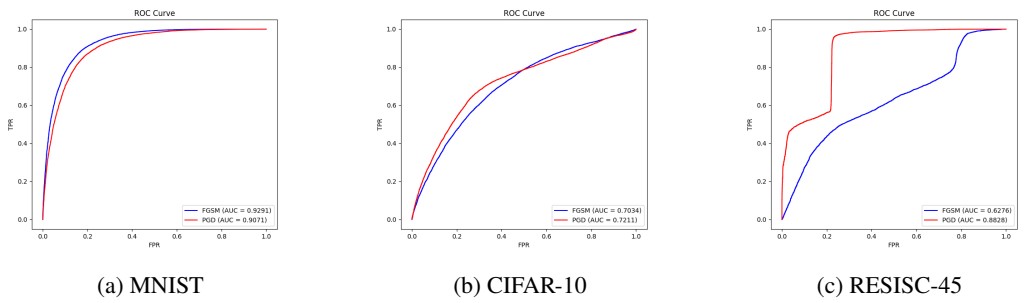

(a) MNIST

(b) CIFAR-10

(c) RESISC-45

Figure 6: ROC curves for adversary detection using uncertainty distribution. The perturbation size $\epsilon$ is 0.3, 4.0/255 and 8.0/255 for MNIST, CIFAR-10 and RESISC-45 datasets, respectively.

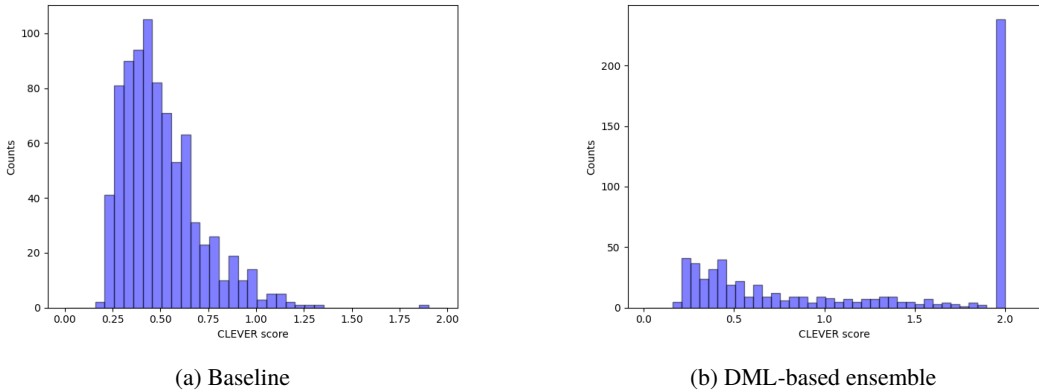

(a) Baseline          (b) DML-based ensemble

Figure 7: Distribution of local CLEVER score for baseline ensemble and DML-based ensemble for 1000 randomly selected images from CIFAR-10 dataset with a CLEVER score of higher than the threshold 0.2. The CLEVER scores are computed on the $L_2$-norm ball with radius 2.0.

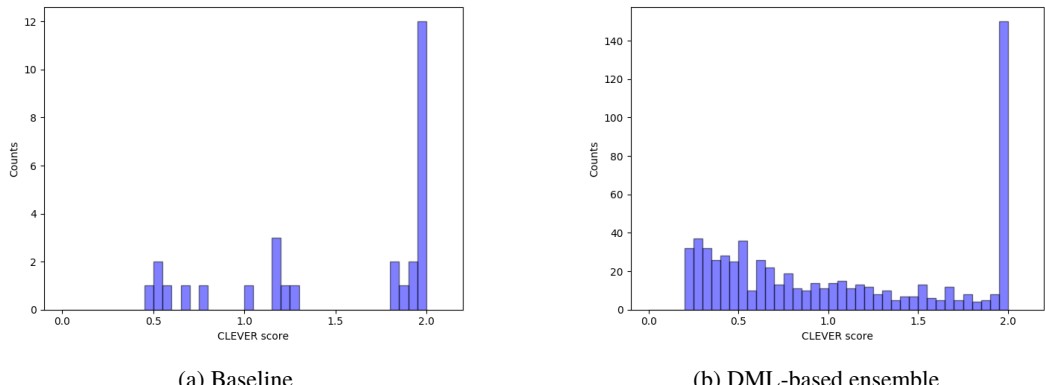

(a) Baseline          (b) DML-based ensemble

Figure 8: Distribution of local CLEVER score for baseline ensemble and DML-based ensemble for 1000 randomly selected images from RESISC-45 dataset with a CLEVER score of higher than the threshold 0.2. The CLEVER scores are computed on the $L_2$-norm ball with radius 2.0.

## A.6 CLEVER SCORE

In Figures 7 and 8 we depicted the histogram of CLEVER score (Weng et al., 2018) of 1000 randomly selected images of the model derived by summing the prediction of individual constituent models in the ensemble for CIFAR-10 and RESISC-45 datasets. The results is shown for samples that have higher CLEVER score than the threshold 0.2. CLEVER score provides an upper bound for the radius of certified ball around each sample in $L_2$ norm. The figures show that the scores from DML-based ensemble outperform of the counterpart of baseline model. For instance for CIFAR-10 dataset the number of samples with the CLEVER score greater than 1.0 is less than 15 samples, however for DML-based ensemble around 250 samples have the score of 2.0.

## A.7 ORTHOGONALITY OF INPUT-OUTPUTS GRADIENTS

Now we demonstrate the effect of pairwise orthogonality of the $M$ matrices for DMLs across the ensemble imposing by regularized $\gamma$ in (3) on orthogonality of the input-output gradients of constituent models. For this purpose we design an ablation study for a ensemble of two models. Once we train the ensemble model with $\gamma = 0.0$ and once the ensemble model is trained with $\gamma = 1.0$. The histograms for the cosine similarity of input-output gradients of the two models in the ensemble over three datasets is shown in Figures 9, 10 and 11. For each dataset we have shown the cosine similarity of models' gradients and also for the sign of gradients. Iterative attacks like PGD apply the sign of gradients in order to craft the attacks. The figures show that the gradients for the models were

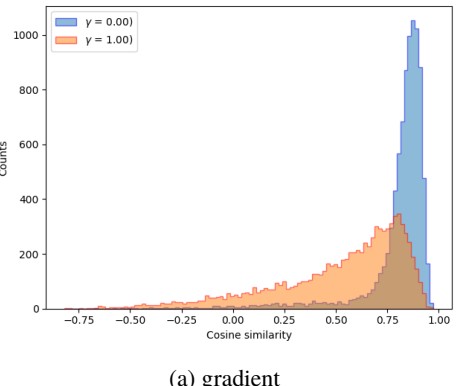
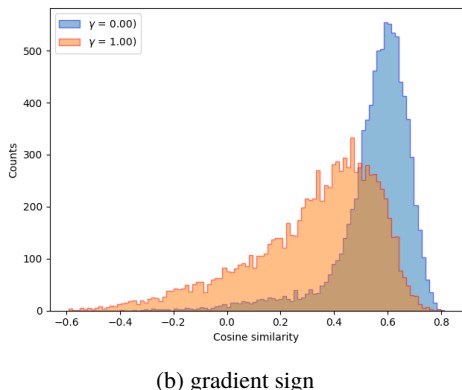

(a) gradient                       (b) gradient sign

Figure 9: Distribution of the cosine similarity of the gradients with respect to inputs for two models in the DML-based ensemble trained with the regularizer term for imposing orthogonality of the $M$ matrices ($\gamma = 1.0$) and the DML based ensemble trained without regularizer term ($\gamma = 0$) for MNIST dataset.

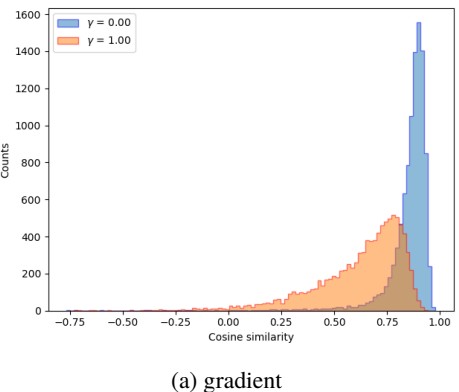
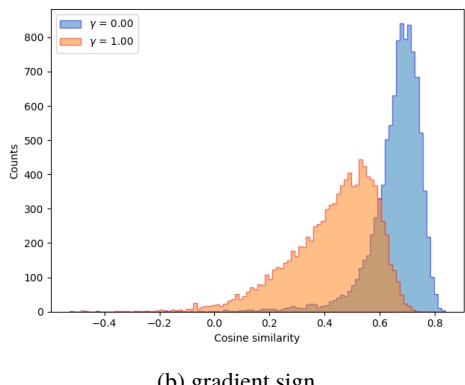

(a) gradient                       (b) gradient sign

Figure 10: Distribution of the cosine similarity of the gradients with respect to inputs for two models in the DML-based ensemble trained with the regularizer term for imposing orthogonality of the $M$ matrices ($\gamma = 1.0$) and the DML based ensemble trained without regularizer term ($\gamma = 0$) for CIFAR-10 dataset.

trained with $\gamma = 0$ are more aligned compared to the ones with $\gamma = 1$. The results for RESISC-45 is not prominent as the cosine similarity of the gradients of ensemble models trained with $\gamma = 0.0$ is already close to zero. The results from these figures confirm that counting on the regularizer in the loss function to impose the orthogonality of the $M$ matrices of paired DMLs would significantly reduce the cosine similarity between the gradients of loss function with respect to the input data for paired models in the ensemble.

## A.8 Effectiveness of R-DML in adversarial training

In this section we illustrate empirically that R-DML can provide better benign and robust accuracy when compared to standard DML for adversarial training. To better distinguish the performance of R-DML from the randomness obtained by augmenting data with Gaussian noise, in addition we also perform experiments for DML where the input data is augmented with the isotropic Gaussian noise, i.e.,

$$g(x) = \mathbb{E}_{\delta \sim \mathcal{N}(0,\sigma^2 I)}(x + \delta - c_f)' M (x + \delta - c_f)$$

We call this layer augmented DML (AUG-DML), as the input to DML augmented with random noise. In Figure 12 we have shown the benign and robust validation accuracy. The validation set is equal to the testset. Figure 13 also show the same counterparts for DML-based models which are obtained

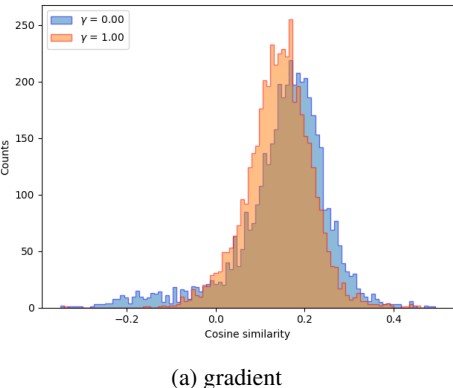
(a) gradient

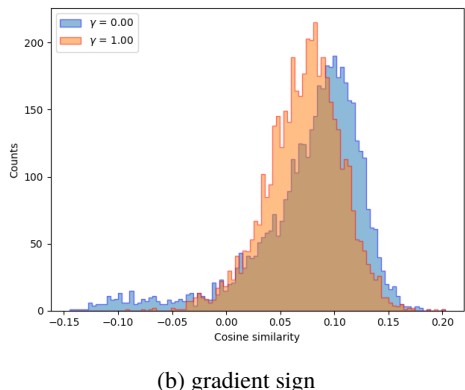
(b) gradient sign

Figure 11: Distribution of the cosine similarity of the gradients with respect to inputs for two models in the DML-based ensemble trained with the regularizer term for imposing orthogonality of the $M$ matrices ($\gamma = 1.0$) and the DML based ensemble trained without regularizer term ($\gamma = 0$) for RESISC-45 dataset.

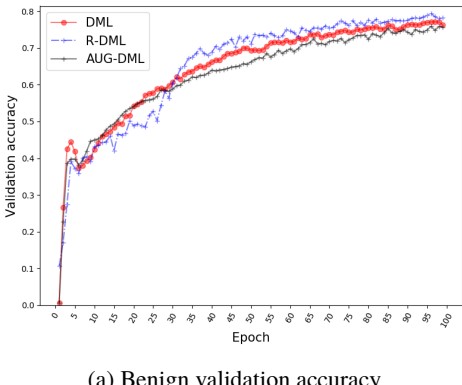
(a) Benign validation accuracy

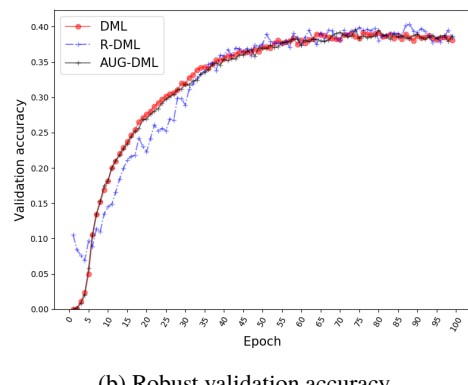
(b) Robust validation accuracy

Figure 12: Benign and adversarial accuracy for adversarialy train models with FGSM attack without random initialization.

by combining DML with the models which are adversarialy trained. In other words for the models in Figure 13 we use prior models which are adversarialy trained. All the models we consider in this section were trained adversarialy with one-step FGSM attack with the stepsize 10.0/255 and without random initialization. The results in Figure 12 show that R-DML marginally show better benign and robust accuracy, while the results from AUG-DML present similar performance as DML. It shows that the noise injected to DML by convolving Gaussian distribution with the $M$ matrix is more effective compared to the DML obtained by augmenting inputs to DML with random noise. Also the results from Figure 13 for R-DML show a significance improvement over DML and AUG-DML. It is important to note that with the higher number of epochs in contrary to DML, R-DML avoids over-fitting. It is also seen that standard DML based network achieve the peak of benign and robust accuracy with only 2 epochs, although the results underperform the ones from R-DML after in the long run training. It shows the DMLs can be applied effectively with pretrained adversarialy trained priors to generate diversity across the model members.

## A.9    RANDOMIZED SMOOTHING AND CERTIFIED ACCURACY

In this section we examine the randomized smoothing model obtained by convolving the base model with the isotropic Gaussian noise (Cohen et al., 2019). Our hypothesis is that the certified accuracy for smoothed classifier with R-DML is providing tighter certified accuracy versus the standard DML. The reason is that R-DML explores some points in the embedding space of the classifier for certifying the samples which might be overlooked in the standard DML case when the input data is augmented

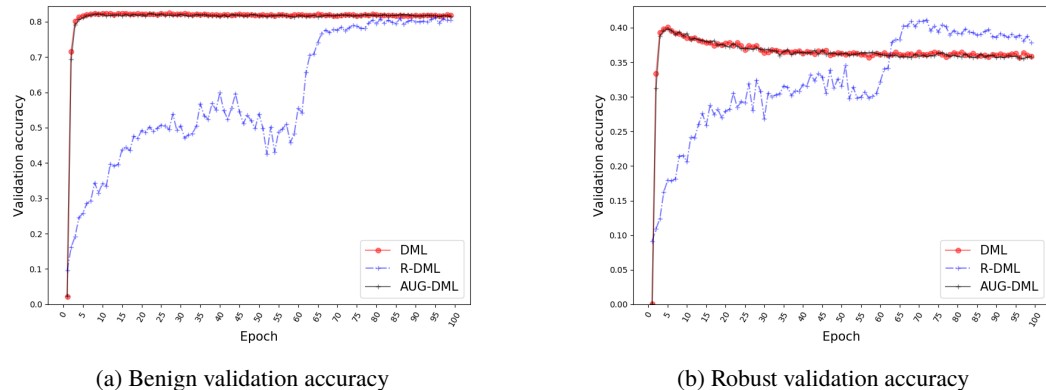

(a) Benign validation accuracy

(b) Robust validation accuracy

Figure 13: Benign and adversarial accuracy for adversarialy train models with FGSM attack without random initialization. The DML based models formed with priors which are adversarialy trained.

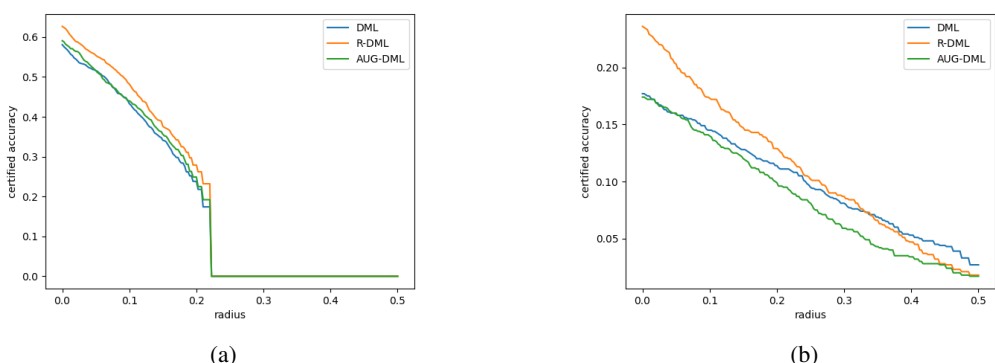

(a)

(b)

Figure 14: Certified accuracy for three types of DML-based model trained standardly. The standard deviation to compute certified accuracy is (a) $\sigma = 0.1$ and (a) $\sigma = 0.25$. The standard deviation for the Gaussian noise of R-DML and AUG-DML is $\sigma = 0.5$.

with Gaussian noise. The reason for that is since we apply DML as a last hidden layer, augmenting features with random noise could add more control on exploration of embedding space. Given that, those embedding points may have different classification as the original sample. Thus, R-DML results in a tighter bound for the certified accuracy by DML. We also demonstrate that AUG-DML defined in the last section does not lead to dramatic change in the certified accuracy compared to R-DML, which supports the significance of applying R-DML. Similar ideas on inserting noise to the embedding space to explore overlooked regions based on data augmentation has been recently explored in for example (Chen et al., 2020). However in this work, the noise is backpropagated in the network to find the mapped location for the noise in the input space, which add additional computational burden to the training procedure. However, in R-DML the noise in embedding space freely added by convolving it to the $M$ matrix. In Figure 14 we have shown the certified accuracy for R-DML versus DML and AUG-DML. Note that these models are trained regularly and not through randomized smoothing training procedure. The random noise for training R-DML and AuG-DML which is applied is isotropic Gaussian noise with standard deviation of 0.5. The certified accuracy for models in this section is measured over 1000 randomly selected samples. Figure 14 shows that R-DML basically provides better certified accuracy compared to the other two DML based models.

In Figure 15 we have shown the certified accuracy of smoothed classifier based on standard DML. The figure also shows the certified accuracy of smoothed classifier, when the DML in the smoothed classifier is replaced by R-DML and AUG-DML. This figure shows that R-DML provides tighter bound for the certified accuracy of smoothed classifier by DML.

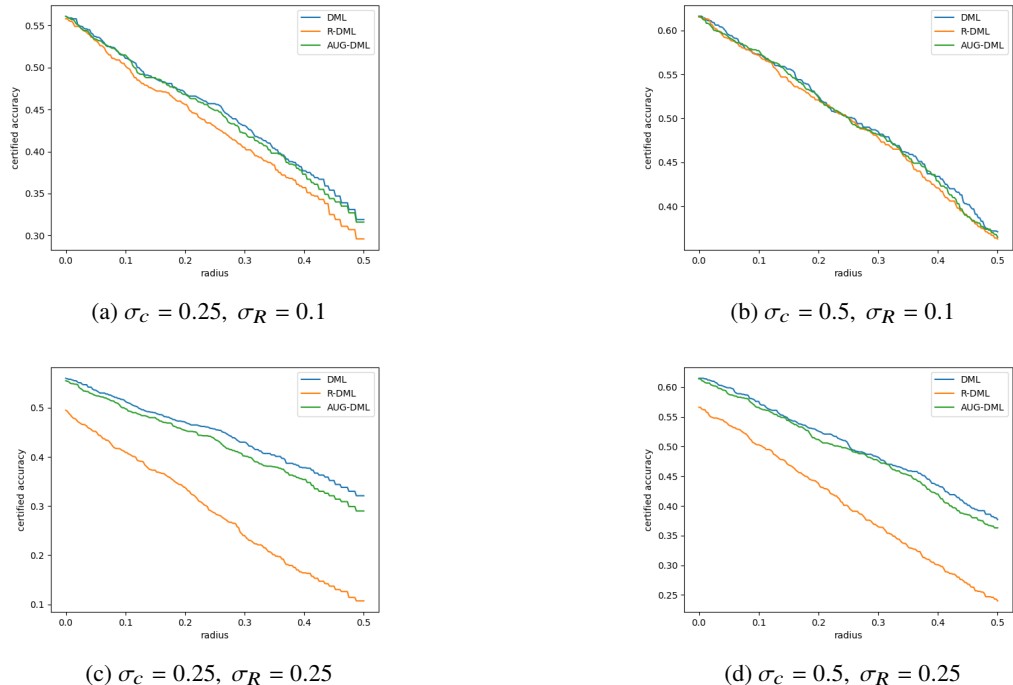

(a) $\sigma_c = 0.25$, $\sigma_R = 0.1$

(b) $\sigma_c = 0.5$, $\sigma_R = 0.1$

(c) $\sigma_c = 0.25$, $\sigma_R = 0.25$

(d) $\sigma_c = 0.5$, $\sigma_R = 0.25$

Figure 15: Certified accuracy for smoothed classifier obtained by randomized smoothing of the classifier with DML. The curves corresponding to R-DML and AUG-DML show the certified accuracy when the DML layer replaced by R-DML and AUG-DML where the std of Gaussian noise is $\sigma_R$. $\sigma_c$ denotes the std of Gaussian noise used to compute certified accuracy.

Table 10: RESISC-45 extensive white-box attack results by scaling the $M$ matrix. FGSM and PGD are applied with specific perturbation scale $\epsilon$. CW is used with the specified penalty parameter $\gamma$.

| Training Model | Benign Acc | FGSM $\epsilon = 0.015$ | FGSM $\epsilon = 0.03$ | BIM-10 $\epsilon = 0.015$ | PGD-10 $\epsilon = 0.015$ | CW $\gamma = 1$ |
|---|---|---|---|---|---|---|
| Baseline | 85.66 | 3.75 | 1.34 | 0.00 | 0.00 | 45.22 |
| DML Ensemble | 88.71 | 44.40 | 33.24 | 23.93 | 25.76 | 36.12 |

In Figure 16 we have shown the certified accuracy for smoothed classifier with three types of DML layer. We then removed the DML layer and replaced it with R-DML to measure the certified accuracy. It is seen that the model trained with AUG-DML does not not tolerate the noise from R-DML and it failed to provide certification for any sample. The smoothed classifier trained by R-DML could provide high certified accuracy.

### A.9.1 EFFECT OF META-MODEL ON PROMOTING DIVERSITY

To show the effectiveness of group-splitting mechanism discussed in earlier, we trained the RESISC-45 dataset without this mechanism and only by scaling the $M$ matrix to impose diversity across members. Figure 17 shows that under this experiment setting, no remarkable diversity in the shape of loss function across members are created. The corresponding results for the robust accuracy and rate of transferability as presented in Tables 10, 11, 12 show lower performance compared to the model trained with meta-model description.

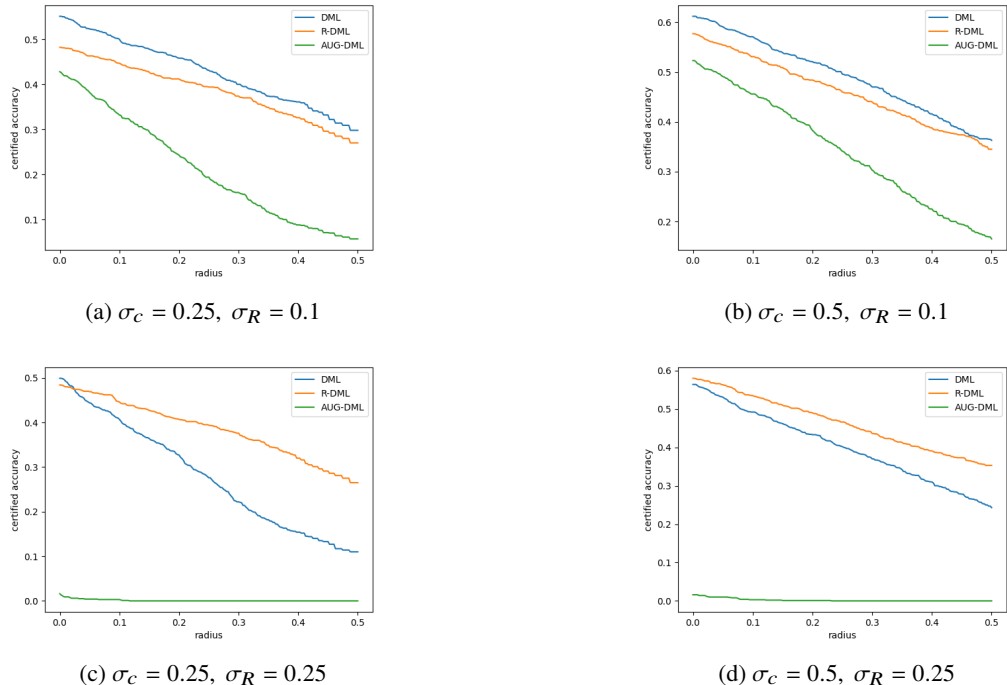

(a) $\sigma_c = 0.25$, $\sigma_R = 0.1$        (b) $\sigma_c = 0.5$, $\sigma_R = 0.1$

(c) $\sigma_c = 0.25$, $\sigma_R = 0.25$        (d) $\sigma_c = 0.5$, $\sigma_R = 0.25$

Figure 16: Certified accuracy for smoothed classifier obtained by randomized smoothing of the classifiers with three types of DML: DML, R-DML and AUG-DML. To calculate the certified accuracy the DML layer for each smoothed classifier is replaced with R-DML where $\sigma_R$ denotes the std of Gaussian noise for R-DML. $\sigma_c$ denotes the std of Gaussian noise used to compute certified accuracy.

Table 11: Adversarial accuracy of models when the attacks are crafted on the target **Model 1** with PGD-10. The labels form the targeted attacks are selected randomly and different from true labels. The size of perturbation $\epsilon$ is 0.015. The model trained without using meta-model and diversity imposed only be scaling the $M$ matrix per each ensemble member.

| **Dataset** | Ensemble | Model 1 | Model 2 | Model 3 | Model 4 | Model 5 |
|---|---|---|---|---|---|---|
| **RESISC-45** | | | | | | |
| Benign Acc | 88.71 | 83.80 | 86.42 | 85.40 | 87.69 | 84.80 |
| Untargeted | 67.96 | 20.40 | 68.22 | 64.91 | 70.31 | 69.21 |
| Targeted | 79.16 | 21.36 | 77.73 | 74.96 | 79.40 | 77.24 |

### A.9.2 DISTORTION-ACCURACY PLOTS

In Figure 18 the distortion-accuracy plots are shown for FGSM, BIM and PGD white-box attacks and over all three datasets. It is observed that increasing the distortion scale $\epsilon$ leads to almost 0% accuracy for the ensemble models which could be an indication that the DML-based models do not mask the gradient information, i.e., the attackers are applying interpretable gradients from the model.

In Figures 19-21 we visualize samples of original images and the corresponding adversarial images generated from Table 5 for all three datasets. These figures show that the DML-based ensemble largely increases the required distortion such that the distortion is highly perceptible.

Table 12: The rate of required $L_2$ distortion for adaptive CW-$L_2$ attack (He et al., 2017) with $\gamma = 1$ for the combined and individual ensemble members of size 5 for 100 randomly selected images. The number of iterations and stepsize are 1000, 0.1, respectively. $\kappa$ is denoting the confidence parameter for CW-$L_2$ attack. The model trained without using meta-model and diversity imposed only be scaling the $M$ matrix per each ensemble member.

| **Dataset** | **$L_2$ distortion required for generating adversaries on each model** | | | | | | |
| --- | --- | --- | --- | --- | --- | --- | --- |
| | Ensemble $\kappa = 0$ | Ensemble $\kappa = 0.1$ | Model 1 | Model 2 | Model 3 | Model 4 | Model 5 |
| **RESISC-45** | | | | | | | |
| Adversarial accuracy | 70.00 | 2.00 | 0.00 | 0.00 | 0.00 | 0.00 | 0.00 |
| Average distortion | 3.69 | 182.90 | 2.77 | 2.72 | 2.73 | 2.82 | 3.32 |

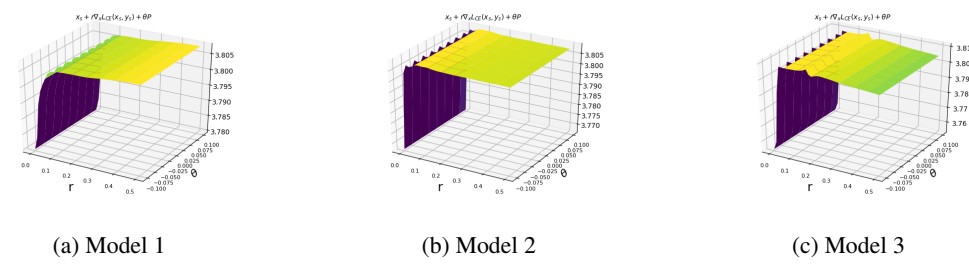

(a) Model 1        (b) Model 2        (c) Model 3

Figure 17: Loss surface for three individual classifiers from the ensemble model without group-wise splitting mechanism. The loss is depicted around the sample $x_s$ with the label $y_s$ from RESISC-45 dataset. The noise vector $P$ is drawn from the normal distribution $\mathcal{N}(0, 0.05)$. The loss surfaces does not show significant diversity across members.

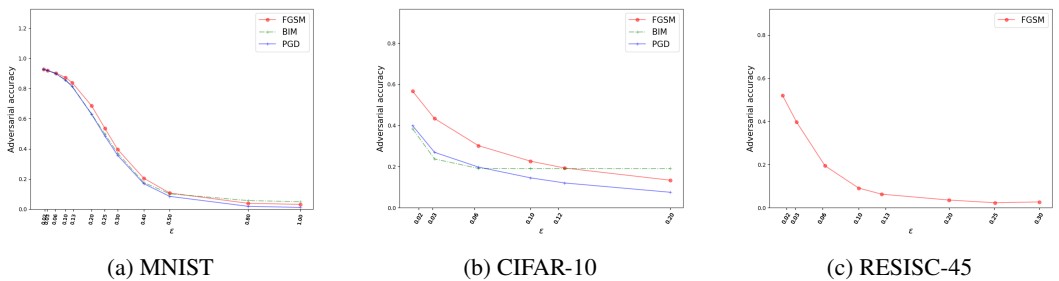

(a) MNIST        (b) CIFAR-10        (c) RESISC-45

Figure 18: Distortion-accuracy curves for adversarial examples crafted by FGSM and PGD white-box aattacks over three datasets.

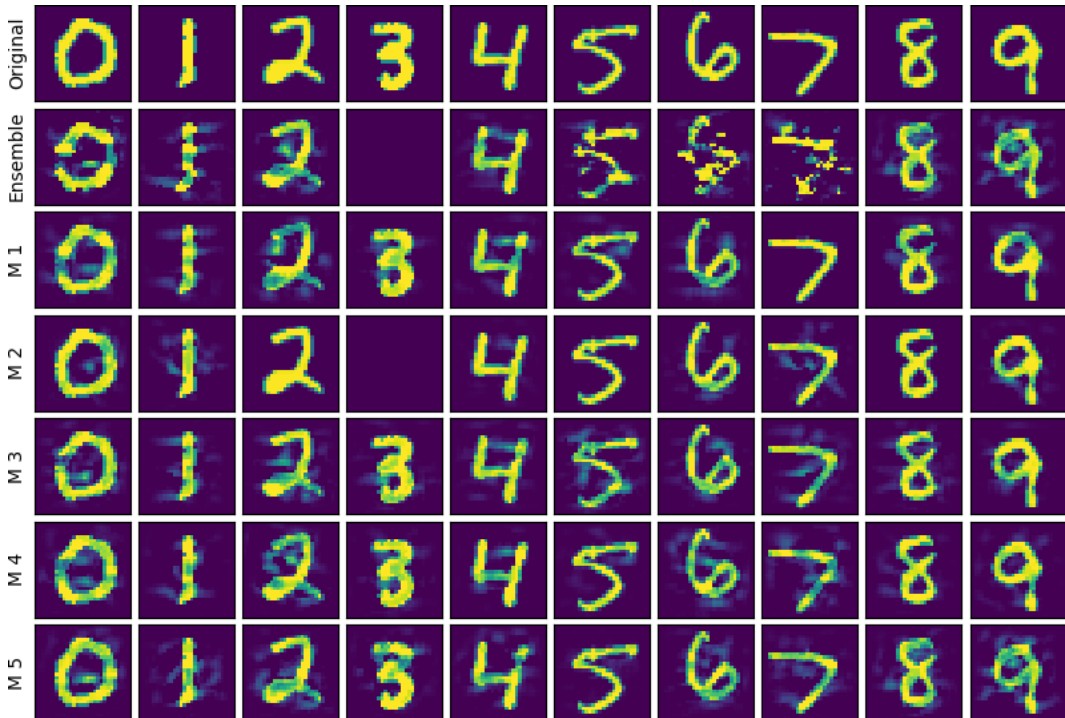

Figure 19: Adversarial samples generated by CW-$L_2$ attack ($\kappa$=0.0) on the original samples from MNIST dataset (first row), and on Ensemble model (2nd row), Model 1 (3rd row), Model 2 (4th row), Model 3 (5th row), Model 4 (6th row), Model 5 (7th row).

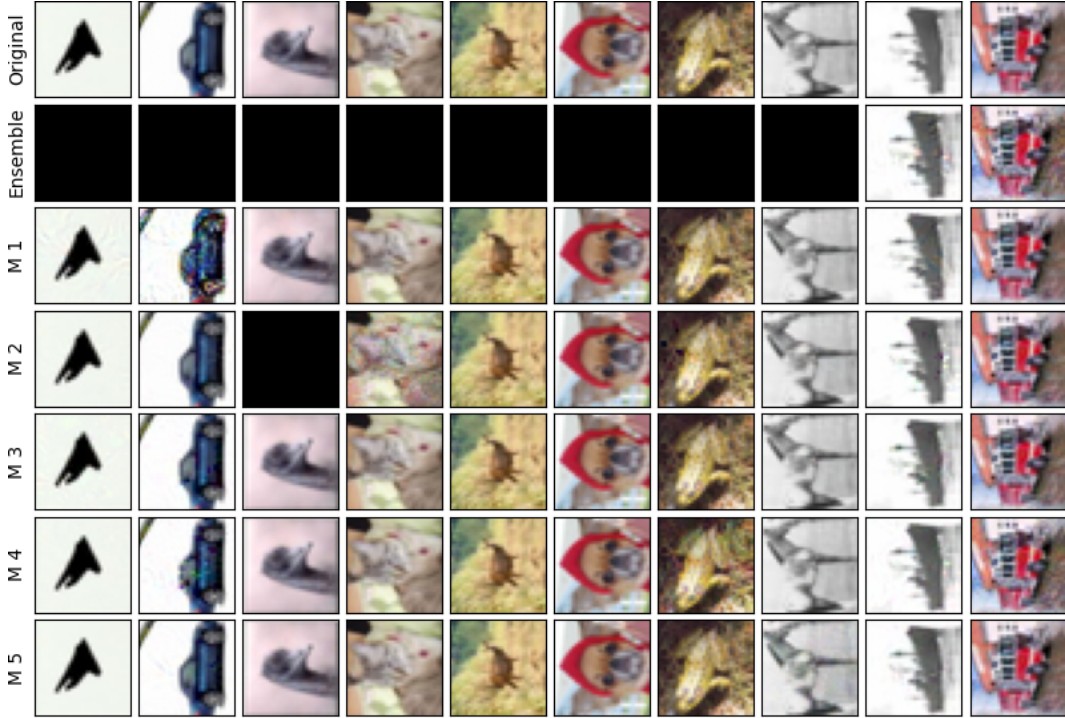

Figure 20: Adversarial samples generated by CW-$L_2$ attack ($\kappa = 0.10$) on the original samples from CIFAR-10 dataset (first row), and on Ensemble model (2nd row), Model 1 (3rd row), Model 2 (4th row), Model 3 (5th row), Model 4 (6th row), Model 5 (7th row).

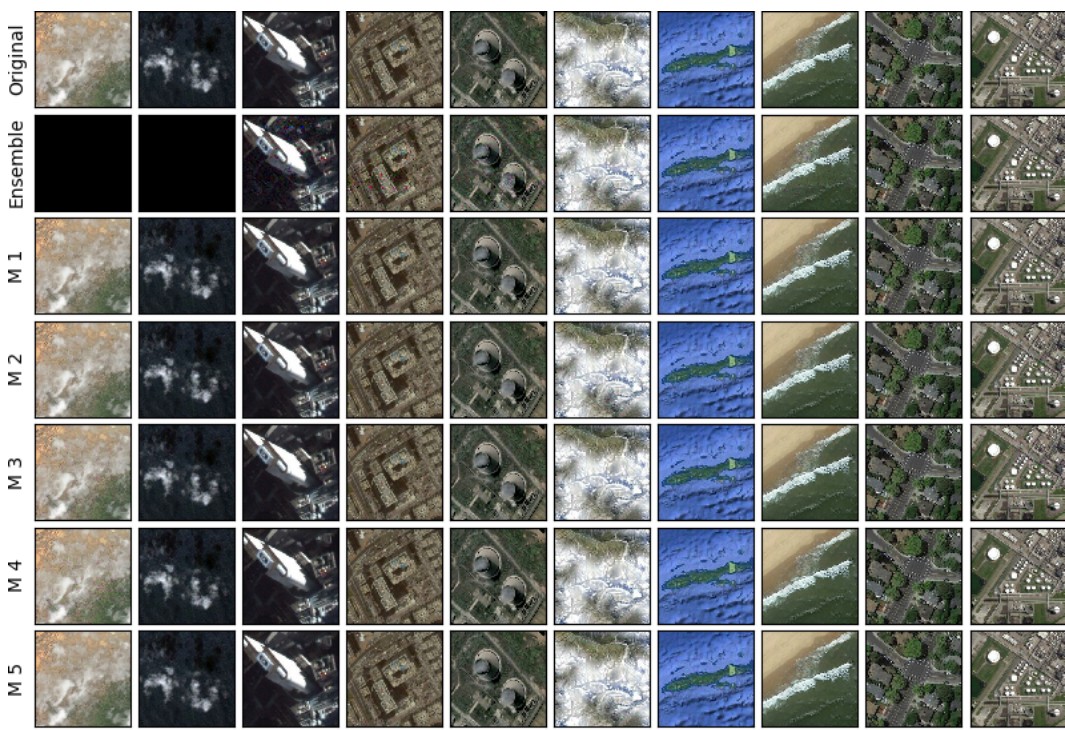

Figure 21: Adversarial samples generated by CW-$L_2$ attack ($\kappa = 0.10$) on the original samples from RESISC-45 dataset (first row), and on Ensemble model (2nd row), Model 1 (3rd row), Model 2 (4th row), Model 3 (5th row), Model 4 (6th row), Model 5 (7th row).

