# OpenReview forum: "Ensemble-based Adversarial Defense Using Diversified Distance Mapping"
_ICLR.cc/2021/Conference — Reject_

### Official Review · AnonReviewer3 · 2020-10-13
**This paper proposes to use the distance map layers for ensemble-based defense**

**Rating:** 4
**Confidence:** 3

**Review:**

Post-rebuttal:
My concerns are not addressed, so I would like to keep my original score.
-------------------------------
This paper proposes to use the distance map layers for ensemble-based defense. By randomly choosing the centers to vary over classifiers, and imposing the I-covariance matrices to be dissimilar and possibly orthogonal, the distance map layers can be used to reduce the rate of transferability of adversarial attacks across models in the ensemble. This paper also demonstrates that distance map layers provide spontaneous regularization of the Lipschitz constant, and therefore further boost the robustness.

I like this paper, but I have some comments as follows:
1. The matrix M in Eq(1) can be represented as M = L^T L. The Mahalanobis distance is equal to
‖x_1-x_2 ‖_M =√((x_1-x_2 )^T LL^T (x_1-x_2))
It denotes that the distance map layers are equal to the fully connected layers. I wonder whether the similar modifications on the fully connected layers can achieve similar performance. If not, please give the reason.
2. There are some proposed SOTA methods like [1], Please follow the experimental settings as [1] for a fair comparison.
3. I suggest the authors to pass this paper through someone that can do the edits to fix the grammatic and typo errors that seem glaring. For example, in eq(1)  ‖x_1-x_2 ‖_M^2->‖x_1-x_2 ‖_M ; in line 4 of abstract, transferablility->transferability …
Thus, I tend to reject this paper.

[1] Tianyu Pang, Kun Xu, Chao Du, Ning Chen, and Jun Zhu. Improving adversarial robustness via promoting ensemble diversity. In ICML, 2019.

---

### Official Review · AnonReviewer2 · 2020-10-27
**This work encourages diversity in ensembles of neural networks through distance map layers (DMLs).**

**Rating:** 5
**Confidence:** 3

**Review:**

This paper aims to improve the robustness of ensembles of neural networks to adversarial attacks. If the models in the ensemble are susceptible to similar attacks, then the ensemble is also vulnerable. This has been shown to be true in practice (He et al.). The authors propose appending distance map layers (DMLs) to the learned representations (i.e., in place of logits) of the member models. The DMLs measure distance to randomly initialized centers using a learned Mahalanobis distance metric. A fraction of the distance matrix, M, is learned such that member M's are approximately mutually orthogonal (thereby decorrelating predictions), and the entries have low L1-norm. The member models' top predictions for each sample are then aggregated by majority vote. The introduction of DMLs is shown to improve robustness to adversarial attacks without a large degradation in accuracy on benign samples. Randomizing the distance metric a bit helps with robustness as expected but hurts benign accuracy also as expected.

The idea of learning orthogonal distance matrices is interesting. I would have liked to see experiments comparing gamma=0 to gamma>0 (I could not find what value of gamma was used in the experiments) to better understand its impact. Even so, I don't understand how orthogonal matrices could be learned effectively with only two free parameters per 10 x 10 matrix.

The distance matrices are learned via backprop on equation (3). Alternatively, the ensemble approach taken appears very similar to a more conventional approach: 1) learn a metric space over the feature representation (i.e., chop off the penultimate layer of the DNN) assuming KNN (k=1); 2) take a majority vote over each model's prediction. See "Distance Metric Learning for Large Margin Nearest Neighbor Classification" '09 JMLR for how to accomplish 1).

The authors emphasize that a low Lipschitz constant is important for improving robustness, but I don't see how it matters if the ensemble is aggregating with majority vote over each member's top prediction.

This peculiarity aside, there are issues with the comparison of the Lipschitz constant for the DML-ensemble as a sum of member models vs the baseline ensemble. 8/10 entries on the diagonal of the distance matrices are fixed to 0.1, so I would expect the Lipschitz constant to be reduced by a factor of about 10 without any learning. This could also be easily accomplished by using the member models as originally intended and introducing a temperature into the softmax (divide logits by 10). The text claims that Figure 1 reveals the mean of the baseline is 3 orders of magnitude larger than the DML-ensemble; by inspection, this looks more like 1 order of magnitude and can be explained by the fact that most of the entries in the matrices are fixed to 0.1.

What $\kappa$ is used for the CW-$L_2$ attack on the individual models in Table 5? Also, only CIFAR-10 shows an order of magnitude difference and that's only if you look at $\kappa=0.1$ despite what's written in the text "order of magnitudes lower than the distortion required to create adversaries".

Why aren't the results for targeted and untargeted the same for Model 1 in Table 4 given that the adversary targets Model 1? Am I misunderstanding the experimental setup?

Model 2 and Model 3 look like they have similar gradients contrary to what's said in the text about Figure 2.

Overall, the direction seems promising, but the paper needs further work.


Quality:
There are some interesting contributions in this paper, but I'm not sure the ultimate model used in the experiments really reveals how important they are. Only learning two entries per distance matrix seems small and I don't see the learned orthogonality reported anywhere. The experimental results are surprising if only a total of 10 additional parameters are learned.

Clarity:
The paper is clear at a high level, however, I had several questions (see above and below) and there were many typos.

Originality:
To my knowledge, learning orthogonal distance matrices for each member in an ensemble is novel.

Significance:
Improving adversarial robustness is an important area of research.


Minor:

In the page 4 Errata note below, you probably mean "\emptyset" $\emptyset$, not $\phi$?

How are the indices of I divided among the ensemble members? Does the 1st ensemble learn M1 = diag([a, b, fixed, fixed, ...]) and the 2nd ensemble member learn M2 = diag([fixed, fixed, c, d, ...]) and so on?

"set of priories" - set of priors

Remember to add parentheses around citations (\citep).

Does I-covariance stand for inverse covariance matrices? Why introduce this new term? Maybe just stick with M if nothing new is being introduced.

---

### Official Review · AnonReviewer1 · 2020-10-28
**Interesting idea, but the presentation and evaluation needs improvements**

**Rating:** 5
**Confidence:** 3

**Review:**

This paper concerns on developing neural network ensembles that can avoid adversarial attacks. The authors propose a new concept of Distance Map Layers (DML) that can be used as the one just before the final layer in a neural network for classification. DML is mainly used to improve the diversity of predictions from the ensemble members, and is defined to be the Mahalanobis distance between an input vector and a output center which may correspond to a class. Each member in the ensemble will learn a different Mahalanobis measure, encoded in the inverse covariance matrix (M), as well as the centers. The training tries to ensure that any two different ensemble members should have inverse covariance matrices of small $L_1$ norms and pair-wise orthogonality, and their set of centers should be far from each other. The authors further propose a randomized DML in which a noise is added to M to help the ensemble to be more robust. To validate the effectiveness of their approach, 3 datasets, 3 network architectures, and 5 adversarial attacks are used in evaluation. Their extensive experiments demonstrate that the DML-based ensemble can perform significantly better than the one without DML, and the randomized DML can perform much better than DML.

Pros:
- The proposed DML is intuitive and promising, especially randomized DML. The idea can be easily used in different contexts.
- The experiments are extensive and really promising.

Cons:
- The notations are inconsistent across the paper. For example, M is used for both dimensionality and inverse covariance matrix. The DML sometimes use squared root (in page 3), but sometimes square (pages 5, 10). Formula (1) is also confusing/wrong.
- The overall presentation is hard to understand and misses many details. For example, there is no section index to refer and follow. Some appendices sometimes are referred without index, causing some difficulties.
- Some formulas needs being revised carefully, e.g., (1), (2), (4), and the proof about the product of two Jacobians in appendix.
- The use of citation should be revised. The authors (quite arbitrarily) mix citations with the main text, which is not a good way.
- It is unclear why one needs some randomness for DML. The authors should provide some reasons behind it. Such a randomness causes some inconsistencies in the obtained Mahalanobis distance, e.g., when $M_f + \delta_i$ is not positive semi-definite. What it means in this case? It is not supportive when the authors say “we apply the uncertainty of Gaussian process to identify the low-confidence regions of input space to the DML layer”. Some theory may be needed.
- When working work with a binary classification problem, the proposed DML-based approach may not provide a good solution. The authors should discuss about this situation and make a comparison with other approaches.
- How to choose/initialize the fixed parts of the inverse covariance matrices. The authors should provide an analysis about this point as it can significantly affect the performance of their method.
- An analysis about the sensitivity of some parameters in their method is also needed, e.g., the effect of the randomness in initialization of the centers, and of $(\lambda, \gamma, \beta, \sigma)$.

---

### Official Review · AnonReviewer4 · 2020-10-28
**Introduces a new technique to design orthogonal ensembles for better adversarial robustness**

**Rating:** 5
**Confidence:** 4

**Review:**

Important: Formatting of section titles - please appropriately number section titles, it makes it very hard to refer to sections otherwise.

The authors address the issue of building robust neural networks by training them using an ensemble based loss function which is designed to promote a sense of orthogonality among ensembles.

The main concern in robust ML is that many attacks are transferable between models, leading to blackbox attacks. This brings the merit of ensemble based methods down -- even though ensembles lead to better accuracies when no attacks are present, attacks which apply to one of the members in the ensemble, can also weaken accuracy of other members in the ensemble and therefore ensembles cannot be stronger against adversarial attacks.

In this paper, the authors derive a new methodology to promote "orthogonality" between different ensembles - which should help reduce the problem of attack transferability. Note that this idea has already been introduced in [1].

The main contribution in this paper is to introduce Distance Map Layers (DML) as an additional layer which maps different class vectors to a class based on their distance from a learnable centroid (each class corresponds to a centroid). This mapping is formulated using Mahalanobis distance, characterized by matrix M. They propose the notion of diversity among ensembles by ensuring that these centroids are at least alpha margin apart for a fixed class, between two different ensembles. This formulation can be seen as an extra regularization term, as can be seen in Eq.(3).

In the experiments section authors demonstrate how when ensembles are trained with this modified loss function, the ensemble accuracy is much better than that of its individual members when attacked by l_\inf or l_2 type attacks.

My main concern is that this isn't the first paper to do ensemble diversity based adversarial training. One of the references also cited in the main text [1]. There are no experimental nor analytical comparisons made to this paper. Until those are made explicit, I'm not sure how the contributions of this paper have an edge over [1].

Please fix typos:
Page 2: "The classification loss function will encourage the distance to the target class to be small while the distances to all other classes to be small, which induces compactness in the embedding space." - other classes to be *large*
Page 4: "we select In's such that Ij \interct Ik " -incomplete sentence ---  = 0.
Page 6: Fig 1 caption: "Distribution lo ..." - Distribution of

References:
[1] "Improving adversarial robustness via promoting ensemble diversity", Pang et. al., ICML, 2019.

---

### Decision · Program_Chairs · 2021-01-07
**Final Decision**

**Decision:**

Reject

**Comment:**

The paper proposes a method to improve adversarial robustness by diversifying the ensemble.

Novelty: As pointed out by several reviewers, promoting diversity of ensembles has been done in the literature, but there's still a moderate novelty in proposing the DML layer.

Empirical validations: The original submission lacks many important comparisons (e.g., with [1]). Despite the authors implicitly compared their method with (Pang et al) via Auto-attack in the rebuttal, it will be better if the comparisons are conducted in a well-controlled way to confirm that the improved robustness comes from DML instead of other hyperparameter settings. Further, it is not clear whether the proposed method is robust to hyperparameters.

Based on these, we recommend rejection but encourage the authors to improve their paper based on the comments.